# Properties of Poultry-Manure-Derived Biochar for Peat Substitution in Growing Media

**DOI:** 10.3390/ma16196392

**Published:** 2023-09-25

**Authors:** Katarzyna Wystalska, Krystyna Malińska, Jolanta Sobik-Szołtysek, Danuta Dróżdż, Erik Meers

**Affiliations:** 1Faculty of Infrastructure and Environment, Czestochowa University of Technology, Brzeźnicka 60A, 42-200 Częstochowa, Poland; katarzyna.wystalska@pcz.pl (K.W.); jolanta.sobik-szoltysek@pcz.pl (J.S.-S.); danuta.drozdz@pcz.pl (D.D.); 2Department of Green Chemistry and Technology, Ghent University, Coupure Links 653, 9000 Ghent, Belgium; erik.meers@ugent.be

**Keywords:** poultry-manure-derived biochar, peat replacement, growing media, horticulture

## Abstract

Peat is considered a contentious input in horticulture. Therefore, there is a search for suitable alternatives with similar properties that can be used for partial or complete peat substitution in growing media. Poultry-manure-derived biochar (PMB) is considered such an alternative. This study aimed at determining the properties of PMBs obtained through pyrolysis at selected temperatures and assessing their potentials to substitute peat in growing media based on the selected properties. The scope included the laboratory-scale pyrolysis of poultry manure at the temperatures of 425–725 °C; the determination of selected physico-chemical and physical properties of the obtained biochars, including the contaminants; and the assessment of the potentials of produced biochars to be used as peat substitutes. PMBs contained less than 36% of total organic carbon (TOC). The contents of P and K were about 2.03–3.91% and 2.74–5.13%, respectively. PMBs did not retain N. They can be safely used as the concentrations of heavy metals, polycyclic aromatic hydrocarbons (PAHs), polychlorinatd biphenyls (PCBs), dioxins, and furans are within the permissible values (except for Cr). Due to high pH (9.24–12.35), they can have a liming effect. High water holding capacity (WHC) in the range of 158–232% *w/w* could allow for the maintenance of moisture in the growing media. PMBs obtained at 525 °C, 625 °C, and 725 °C showed required stability (H/C_org_ < 0.7).

## 1. Introduction

### 1.1. Peat in Horticulture

Peat is one of the most common substrates in horticultural growing media, specifically used in containerized plant production in nurseries and greenhouses to allow optimal development of plants in pots (seedlings, transplants) [1,2]. It is estimated that about 20 million tons of peat are extracted annually in Europe [2]. The main peatlands in Europe are located in Finland, Sweden, Belgium, Estonia, France, Germany, Ireland, Latvia, Lithuania, the Netherlands, Norway, Poland, and the United Kingdom [3]. The use of peat covers 62% for energy and 38% for non-energy purposes, including horticulture. In Europe, peat accounts for up to 80% of the growing media [2,4,5].

Peat is a nonrenewable material that consists of decomposed plant material accumulated in a water-saturated environment and in the absence of oxygen [2,6,7]. It contains various organic components, but chemical composition varies due to complex chemical structure [1]. The characteristics of peat parameters depend on the type of plants, the degree of decomposition of organic matter, and the location and season of harvesting. For example, horticultural peat that is directly collected from peatland shows pH in the range of 2.8–4. To increase pH, peat is often amended with chalk (pH increase to 5–6.5). Peat products that are available on the market can contain a mixture of perlite, compost, sand, or bark that change the physical and chemical properties and thus allow for the diversified use in growing plants [8,9]. As for the content of nutrients, for example “sphagnum” raised peat is characterized by N 0.6–1.2%, P 0.02–0.04%, K 0.01–0.08%, Ca 0.14–0.35%, and Mg 0.06%. However, for low peat, the content of nutrients is N 2.5–3.5%, P 0.08–0.26%, K 0.04–0.17%, Ca 0.21–0.36%, and Mg 0.19–0.31% [10].

The properties of peat that make it a highly efficient component in growing media include its low weight; stability; high water holding capacity; ability to absorb and release nutrients; low pH of 3.5–5; low content of nutrients; low number of pathogens; and lack of heavy metals, potentially toxic substances, and weed seeds. It is recommended that since peat has low contents of nutrients, it needs to be supplemented by adding fertilizers [11]. Peat has been a predominant substrate for growing media not only due to its suitable properties but also due to the fact that peat demonstrates consistency in physico-chemical and physical properties, stability, and availability. Therefore, peat is relatively easy in transportation and storage [12]. 

Despite the number of advantages, peat as a substrate for growing media is considered a very contentious input in horticulture [13,14]. This is mostly due to peat excavation, which has caused drainage of peatlands, and in consequence in many cases has contributed to irreversible damage to the environment. Peat is a significant storage of organic carbon, with an estimated 500 billion to 1 trillion tons of carbon stored in peatlands worldwide. When peatlands are drained, CO_2_ is released into the atmosphere (from 1 to several tons per year). It is reported that at least 15–20% of peatlands worldwide have already been drained, contributing to CO_2_, nitrogen oxide, and methane emissions to the atmosphere. According to some estimates, 1 hectare of peatland can accumulate about 1 ton of CO_2_, while drained peatland emits from a few to many tons of CO_2_ into the atmosphere [15]. The strong pressure from the society and authorities to protect peatlands resulted in changes in legislation. The European Union has been working on legislation to restore degraded ecosystems, including peatlands [7,16]. In the UK, to protect the peatlands the government will introduce a ban from 2024 on selling peat to amateur gardeners [11]. In view to this, there is a strong pressure to reduce the use of peat, primarily in horticultural applications, and replace it with alternative materials. Therefore, especially in recent years, we are observing a global increase in the scale of searching for alternative materials that demonstrate similar properties to peat but are sustainable and circular [12]. Such materials include a wide range of byproducts from industry and different types of biomass and agricultural residues. For example, materials such as coir, sheep manure, soil and perlite, bark, wood fibers, and green waste compost have been tested in horticultural applications as peat alternatives [7,11,17,18]. All of these materials for peat substitution are expected to demonstrate similar properties, in particular such physical properties as water-holding capacity, bulk density, air-filled porosity, and structural stability. Such materials should be easily modified, e.g., to obtain the appropriate pH and/or nutrient content. What is more, potential peat alternatives should demonstrate low susceptibility to degrade during plant growth and should be free from contaminants and weed seeds [14]. It is also required that peat alternatives should be inexpensive and sustainable [12].

### 1.2. Biochars as Substrate for Peat Substitution

There have been a number of studies that investigated biochars as suitable alternatives to replace peat from horticulture [19]. This is mostly due to the fact that biochar parameters can be easily modified, in terms of a substrate, pyrolysis temperatures, and heating time [20,21,22]. Biochar is also more resistant to microbial activity, so it can have longer viability as a growing medium or as an addition to peat [4]. However, biochars can differ in their properties depending on a substrate used for pyrolysis, chemical composition, particle size, bulk density, water holding capacity, etc., and therefore there is a need to have better understanding of how plant-derived and manure-derived biochars perform as peat substitutes. Biochar, compared to peat, demonstrates high adsorptive properties of heavy metals and has the effect of reducing the salinity of the soil, which results in reduced stress for plants. It is also noted in the literature that biochar has a positive effect on soil properties and the growth and yield of plants [23].

Recent studies on biochars for peat substitution report the results on the use of mostly plant-derived biochars as a partial substitute or a complete substitute of peat in growing media [19,24]. There is little known about biochars from poultry manure used as a partial or complete substitute for peat in growing media. 

### 1.3. Poultry-Manure-Derived Biochars

Undoubtedly, biochars have been extensively studied in terms of properties and applications. This also refers to poultry-manure-derived biochars. These properties can vary significantly due to the use of different types of poultry manures (i.e., feeding, type of breeding system, etc.) and the pyrolysis parameters (i.e., the process temperature, heating and holding time). The pyrolysis parameters allow for tailoring chemical and physical properties of biochars such as chemical composition, specific surface area, porosity, and presence and type of functional groups. These properties of biochars determine their applications. Generally, biochars produced from poultry manure demonstrate alkaline pH (on average in the range of 7–12); total carbon content less than 50%; nitrogen content of about 1 to 5%; and the presence of Ca, Mg, P, and K (Table 1). 

Although poultry-manure-derived biochars have been studied by many researchers, it has to be pointed out that the literature does not provide sufficient information on the properties of poultry-manure-derived biochars (i.e., missing data including parameters such as bulk density and water holding capacity) in the context of peat substitution and phasing out peat from horticulture. In addition, there is little information on the potential risks related to the contaminants that can be present in such biochars such as heavy metals, dioxins and furans, polycyclic aromatic hydrocarbons (PAHs), and polychlorinted biphenyls (PCBs). The occurrence of these contaminants in biochars can depend on some feedstock parameters such as moisture content as well as the process parameters. Polycyclic aromatic hydrocarbons belong to the group of persistent organic pollutants (POPs) with ≥2 aromatic rings. They tend to persist in the environment and as such can pose carcinogenic, mutagenic, and teratogenic effects on living organisms [39,40]. Recent studies reported that the formation of PAHs during pyrolysis cannot be easily predicted as it results from the combination of different factors such as feedstock composition, pyrolysis temperature, retention time, and atmosphere [40]. The occurrence of these contaminants in biochars may limit their use for soil applications. 

The overall goal of this study was to determine the properties of poultry-manure-derived biochars obtained through pyrolysis at selected temperatures and to assess their potentials to substitute peat in growing media. The scope of the presented study included: (1) laboratory scale pyrolysis of poultry manure in the selected temperatures (425–725 °C); (2) determination of physico-chemical and physical properties of the obtained biochars, including the contaminants; and (3) assessment of the potentials of produced biochars to be used as a peat substitute. This study contributes to the state of the art as it aims at analyzing the properties and assessing poultry-manure-derived biochar as a substrate for partial or complete substitution of peat. With the knowledge on the required properties for peat substitution, poultry-manure-derived biochar could be used as a partial or complete peat substitute to phase out peat from horticultural applications. 

## 2. Materials and Methods

### 2.1. Poultry Manure for Biochar Production

Poultry manure was sampled from a local poultry farm in Southern Poland (with a cage breeding system), with the average population of 30,000–40,000 laying hens (personal communications). Prior to laboratory pyrolysis, poultry manure was tested for pH, moisture content, ash, total carbon, total nitrogen, and phosphorus. The properties of the poultry manure used for our studies on converting poultry manure into biochar within the Nutri2Cycle project varied slightly. On average, as reported in our previous work, the moisture content was about 80%, organic matter—75% (d.m.), organic carbon—43% (d.m.), nitrogen—8% (d.m.), total phosphorus—0.01% (d.m), and pH—7.5. Bulk density (wet) was about 910 kg·m^−3^, whereas air-filled porosity was 20% [30,31,41].

### 2.2. Laboratory Biochar Production

Poultry manure was pyrolyzed in a laboratory pyrolysis reactor (PRW-S100x780/11; manufactured by the Czylok company from Jastrzębie-Zdrój, Poland)) in nitrogen atmosphere (5 L·min^−1^) as reported in our other studies [30,41]. Pyrolysis of poultry manure was performed at selected temperatures: 425 °C, 525 °C, 625 °C, and 725 °C. The heating times were 120 and 150 min for the temperatures 425 °C, 525 °C, and 625 °C, and 725 °C, respectively. The holding time was 60 min for all selected temperatures. After the completion of the process, the pyrolyzed samples were left in the reactor to cool down. The produced biochars (in this paper referred to as PMB 425 °C, PMB 525 °C, PMB 625 °C, and PMB 725 °C) were crushed in a laboratory mill and were graded by an electromagnetic sieve shaker (AS 300, Control, RETSCH, Haan, Germany), and stored in sealed containers [42]. 

### 2.3. Analysis of Biochar Properties

The obtained poultry-manure-derived biochars were subjected to the analyses of physico-chemical and physical properties. pH was measured in distilled water; ash was determined according to the standard PN-EN ISO 18122:2016-01 Polish version [43], Solid biofuels—Determination of ash content; bulk density (BD) was determined according to the PN-EN 1236 standard [44] on fertilizers; and water holding capacity (WHC, % *w*/*w*) was determined from the difference in mass between dry and saturated biochar according to ASTM D2216-10 [45]. Selected elements including P, K, Ca, Mg, Hg, Pb, Cd, Cr, Cu, Ni, and Zn were determined in the investigated biochars with the use of ICP-OES according to the standard PB-186/ICP. Detailed methods for analyzing biochars are provided in our previous works [30,31,41]. Total carbon (TC) was determined by Multi N/C, Analytykjena—the high-temperature incineration with detection IR according to PN-ISO 10694:2002 [46]—Soil quality—Determination of organic carbon content and total carbon content after dry combustion (elemental analysis). The elemental analysis of CHNS was performed according to the standards CSN ISO 29541 [47], CSN EN ISO 16994 [48], CSN EN ISO 16948 [49], CSN EN 15407 [50], CSN ISO 19579 [51], CSN EN 15408 [52], and CSN ISO 10694 [53]. BET surface area was determined by nitrogen sorption by the ASAP2020 Plus analyzer (Micromeritics, Atlanta, GA, USA). The investigated poultry-manure-derived biochars were tested for the presence of dioxins and furans: US EPA 1613B [54] and CSN EN 16190 [55], and polycyclic aromatic hydrocarbons (PAHs) and polychlorinated biphenyls (PCBs): US EPA 8270D [56], US EPA 8082A [57], CSN EN 15527 [58], ISO 18287 [59], ISO 10382 [60], CSN EN 15308 [61], and US EPA 3546 [62]. 

### 2.4. Sorption of N_NH4_ and P_PO4_ by Poultry-Manure-Derived Biochars

In addition to these analyses, we also performed sorption of N_NH4_ and P_PO4_ in the lysimetric columns. A sample of biochar (200 cm^3^) was placed in a lysimetric column (diameter of 50 mm and height of 1000 mm), which was filled with 400 mL of the solutions with the targeted concentration of nitrogen (the concentration of N_NH4_ of 50 mg·L^−1^) and phosphorous (the concentration of P_PO4_ of 27 mg·L^−1^), as well as the mixture of nitrogen and phosphorus (in the same concentrations). pH was adjusted to 4.9, as this is the typical pH for the soil solutions from acidic soils that predominate in Poland. The leachates from each column were collected and analyzed for nitrogen and phosphorus [63].

## 3. Results

### 3.1. Biochar Yield

Pyrolysis of poultry manure at the selected temperatures 425 °C, 525 °C, 625 °C, and 725 °C resulted in a biochar yield in the range of 39.59–51.24% (Table 2). It was observed that with the increase in pyrolysis temperature, the biochar yield decreased. This was due to the loss of volatile substances at higher pyrolysis temperatures. 

### 3.2. Physico-Chemical and Physical Properties of Biochars

Selected properties of the investigated biochars are presented in Table 3. The average moisture content of the obtained biochars was about 4%. The particle size of the investigated biochars ranged from 5 mm to 0.063 mm (fractions of 2–5 mm: 4%; 1.6–2 mm: 6%; 1.4–1.6 mm: 6.2%; 1.0–1.4 mm: 15.2%; 0.5–1 mm: 28.2%; 0.25–0.5 mm: 22.6%; 0.1–0.25 mm: 12.8%; 0.063–0.1 mm: 2.4%; 0–0.063 mm: 1.6%). Typically, poultry-manure-derived biochars demonstrated alkaline pH in the range of 9.24–12.35. With higher pyrolysis temperatures, the pH values increased, which is attributed to the increase in alkaline cations (Ca, Mg, K) in the biochars produced at higher temperatures. The ash content of the investigated biochars was high and ranged from 55.61% to 73.34%. This is typical for biochars obtained from animal manures or sewage sludge in contrast to biochars derived from plant biomass. Increased pyrolysis temperature results in the increase in ash due to the loss of volatile substances. Total carbon of PMBs was low and in the range of 31.28–38.57%. Low C content in these PMBs is typical for manure-derived biochars and is significantly lower than in the case of biochars produced from plant biomass in which lignin facilitates carbonization and results in the increase in carbon in the pyrolysis products [36]. Pyrolysis temperature did not have an effect on the contents of total carbon. Similar observations were reported by Roberts et al. (2017) [64], who determined C in biochars produced from biosolids at temperatures of 450 °C, 600 °C, and 750 °C. Significant differences in the content of carbon in the solid pyrolysis products are usually observed in plant-derived biochars. 

The content levels of P and K were about 2.03–3.91% and 2.74–5.13%, respectively. Similar contents of these elements were reported in other studies [25,28,65]. The P and K contents increased with the pyrolysis temperature, which is typically observed in biochars at these temperatures [25,65]. The content of Ca in the investigated PMBs ranged from 12.20% and 16.70%, and it increased with the increase in the pyrolysis temperature. Other researchers reported that the Ca content in the investigated biochars was in the range of 7.17–9.40% and also increased with higher pyrolysis temperatures [28]. As for Mg in the PMBs, it was in the range of 0.83–1.53%. 

The cation exchange capacity (CEC) significantly increased with higher pyrolysis temperature, which was confirmed by other studies. For example, Bavariani et al. (2019) [25] studied the cation exchange capacity of biochars produced from poultry manure, which was in the range of 58.0–86.5 cmol·kg^−1^. 

Poultry-manure-derived biochars demonstrated low surface area compared to, e.g., biochars from woody materials, as reported by other researchers (Table 1), which slightly increased at higher pyrolysis temperature. The surface areas of the investigated PMBs were in the range of 11–18 m^2^·g^−1^. These low values can result from a high content of ash in the biochars, which in turn could block the development [26] of surface area due to clogging of pores by inorganic compounds present in ashes [28,65]. Other researchers reported that biochars produced from poultry manure at temperatures of 400 °C and 600 °C demonstrated BET values at 4.3 m^2^·g^−1^ and 5.34 m^2^·g^−1^, respectively [66]. The BET values of the investigated PMBs were about twofold those reported in the literature. 

Bulk density (wet basis) ranged from 182 to 251 kg·m^−3^. Bulk densities can vary due to moisture content and particle size and distribution. The highest bulk density of poultry-manure-derived biochar was observed for biochar produced at 725 °C. This could have been due to the particle size and distribution of biochar produced at this temperature. It has to be pointed out that poultry-manure-derived biochar is a somewhat heterogeneous material, and as such, the values of bulk densities can vary. The investigated biochars demonstrated high water holding capacity (WHC) (Table 3). Generally, biochar can absorb water up to 5.0 times its own weight [67]. It was observed that biochars with mixed particle size, irregular shapes, and hydrophilic properties were able to rapidly store relatively large volumes of water (up to 400% wt.) [68]. It was pointed out that biochars added to coarse soils could be more efficient in terms of water availability to plants. However, the researchers suggested that there is still little understanding of the effect of biochar on the water holding capacity of soil [69]. The literature does not report any studies on the effect of poultry-manure-derived biochars on the water holding capacity of growing media.

The produced biochars were subjected to the elemental analysis (Table 4). The total organic carbon (TOC) was low and did not exceed 36%. According to the European Biochar Certificate [68], the content of total organic carbon in biochars for horticultural applications should be more than 50% (d.m.). The content of H in the PMBs was in the range of 1.17–2.37% and decreased with higher pyrolysis temperatures due to the loss of -OH groups by biochars, as observed by Bavariani et al. (2019) [25] and Lie and Chen (2018) [70]. A similar tendency was observed for N (3.02–4.00%) that was confirmed in the literature [65,70]. The content of S in the investigated biochars was in the range of 0.30–0.52%. The maximum of the H/C ratio in biochars according to the Regulation (EU) 2019/1009 should not exceed 0.7. This is also in line with the EBC guidelines [71] and the IBI standards [72]. The molar ratio of H/C_org_ is related to thermochemical changes that lead to the formation of condensed aromatic ring structures. These are the indications of biochar stability [71]. In the case of the investigated biochars, the H/C_org_ of PMB425 °C was higher than the recommended value, which can indicate that the pyrolysis temperature was not sufficient to obtain stable biochar. Aromaticity of biochars also affects their sorption properties [73].

The PMBs were tested for the content of heavy metals: Cd, Pb, Cr, Cu, Ni, Zn, and Hg (Table 5). Bioavailability of microelements present in the poultry feed, such as Zn, Cu, Mn, and Fe from inorganic compounds, is estimated at about 30%. The remaining is excreted by poultry with manure in higher quantities, which could be a source of environmental contamination. During pyrolysis of poultry manure, the loss of mass is related to the loss of volatile substances and has an effect on the concentration of heavy metals in the produced biochars. The concentrations of Cd in the PMBs was in the range of <0.300 to 0.610 mg·kg^−1^ and was decreasing with the higher pyrolysis temperatures. The concentrations of Pb did not exceed 2.05 mg·kg^−1^. As for Cr, the concentrations were in the range of 17.10–29.80 mg·kg^−1^. The concentrations of Cu ranged from 84.6 to 158 mg·kg^−1^, whereas of Ni: 16.4–32.8 mg·kg^−1^ and Zn: 434–831 mg·kg^−1^. The concentrations of Cr, Cu, Ni, and Zn in the investigated biochars increased with higher pyrolysis temperatures. The concentration of Hg in all PMBs was low and did not differ significantly with the increase in the temperature. For example, Srinivansen et al. (2015) [36] determined the content of heavy metals in biochar from poultry manure and the concentrations of these metals (Zn—195.4 mg·kg^−1^, Cu—24.77 mg·kg^−1^, Cr—12.25 mg·kg^−1^, Cd—0.28 mg·kg^−1^, Pb—2.31 mg·kg^−1^, Hg—0.02 mg·kg^−1^) did not exceed the limits recommended by the EBC guidelines [71]. The reported concentrations of heavy metals were lower, except for Hg, but the content of heavy metals can vary depending on, e.g., feed type. According to the Fertilizing Product Directive (Regulation (EU) 1009/2019) [74], the concentrations of heavy metals in the investigated PMBs were within the permissible limits, except for Cr. 

The content of dioxins and furans in the produced biochars (Table 6) was low and did not exceed the limit recommended by the ECB and required by the Regulation (EU) 2019/1009 (Annex I), which for PCDD/F is 20 ng·kg^−1^ (I-TEG OMS) [71]. The pyrolysis temperatures used in our study had no effect on the PCDD/F content in the produced biochars. The formation of dioxins is influenced by the presence of chlorine and organic matter in the substrate and the temperature of the process. The presence of chlorine in poultry manure depends on the farming system [75]. The chlorine content of free-range livestock manure is comparable to that of plant biomass, while it is considered too high in industrial manure. As shown by the authors, the chlorine content in free-range litter was 0.11%, while in industrial manure, it was in the range of 0.66–0.99% [75]. Similar results were reported by Quiroga et al. (2010) [76]—chlorine content at the level of 0.64%, and Adamczyk et al. (2021) [77]—1.12%. Dioxins and furans are mainly associated with the combustion process. However, the pyrolysis process is not free from the formation of these substances [78].

The pyrolysis of poultry manure containing organic matter can result in the formation of polycyclic aromatic hydrocarbons (PAHs) [71]. PAHs are usually formed at lower temperatures (<500 °C) during fast pyrolysis and shorter residence time [39]. A recent study reports that there is no safe pyrolysis temperature that would prevent the formation of PAHs, but it was stated that at high pyrolysis temperatures, larger (4–6 rings) and less bioavailable PAHs are formed [40]. The results of the PAH analysis performed for the investigated biochars (Table 7) demonstrated that the sum of 16 PAHs did not exceed the value of 6 mg·kg^−1^ (d.m.) as required in the Regulation (EU) 2019/1009 (Annex I). Also, it did not exceed the value of 0.19 mg·kg^−1^ (d.m.) recommended by the ECB standards. PAHs produced during pyrolysis are usually released with gaseous products, occasionally with the use of inappropriate technological solutions leading to the contamination of biochar with PAHs [71]. These can include, e.g., the occurrence of oxygen during pyrolysis [39]. According to some studies, the contents of PAHs in biochars do not depend on the type of a substrate [79]. However, in the recent study, the authors reported that the content of PAHs in the investigated biochars was not significantly correlated to feedstock or temperature [40]. It has been reported that biochar (with no excessive concentration of PAHs) used in soil does not pose a risk of releasing PAHs but may act like a sorbent rather than a source of PAHs [80,81]. For example, as reported in other studies, adding biochar to sewage sludge and then to soil had an effect on PAH reduction and reduced sewage sludge toxicity [82,83]. 

The analysis of the PCB content in the investigated biochars (Table 7) showed that for biochars PMB425 °C, PMB525 °C, and PMB625 °C, the PCB concentration was < 0.0200 mg·kg^−1^ (d.m.) and did not exceed the limit value recommended by the ECB. However, in the case of biochar PMB725 °C, the PCB content was determined at the level of <0.5270 mg·kg^−1^ (d.m.). This value significantly exceeds the established limit of 0.2 mg·kg^−1^ [71]. However, according to the Regulation (EU) 2019/1009 (Annex I), the content of PCBs should not exceed 0.8 mg·kg^−1^ (d.m.). It has to be pointed out that there is little information on the formation of pyrogenic contaminants in poultry manure and there is a need for better understanding how these contaminants are formed [40,84].

### 3.3. Sorption Properties of Biochars

The results of the sorption test are presented in Table 8. The most preferable sorption properties towards N_NH4_ and P_PO4_ were demonstrated by PMB 725 °C. This type of biochar demonstrated the highest surface area (18 m^2^·g^−1^) and bulk density (251 kg·m^−3^), as well as the lowest microspore area (2.6 m^2^·g^−1^). These could have an impact on water holding capacity. As for PMBs produced at lower temperatures, it was observed that the concentration of N_NH4_ in the initial solution after performing the test increased. This shows that these biochars contain high quantities of soluble nitrogen compounds. In particular, this phenomenon was observed for PMB 525 °C, which indicates its fertilizing potential. In the work of other researchers, it was reported that during the extraction with deionized water, no presence of N_NH4_ was detected [85]. According to Rathnayake et al. (2023) [84], during pyrolysis, nitrogen present in poultry manure is converted into heterocyclic compounds such as pyridinic-N, pyrrolic-N, and graphite-N or quaternary N, which show low bioavailability of N. Therefore, biochars show lower N availability than fresh manure, digested manure, or composted manure. The results of P sorption demonstrated that the removal of the initial concentration of P was above 97% for all PMBs (except from PMB 625 °C). No leaching of phosphorus from the PMBs was observed. Similar results were reported in other studies where the researchers attempted to extract phosphorus with water from biochar produced at 500 °C, resulting in the extract with a P concentration of 1% [86]. As for the initial solutions containing both nitrogen and phosphorus ions, sorption was similar to the single-ion solutions.

According to the literature, the application of biochars to soil can improve the retention of nutrients and reduce the capacity of soil leaching [87]. However, the effect of biochar combined with other additives to soil can vary. Biochars can improve properties of soil with low fertility or high acidity, or can have neutral or even a negative effect on high-fertility soils [88]. The presented results demonstrated that a higher pyrolysis temperature resulted in lower concentrations of nitrogen-soluble forms. The investigated PMBs showed sorption potential towards phosphorus. We did not observe high leaching of phosphorus compounds, which may indicate that phosphorus is present in sparingly soluble forms. However, other studies reported that biochar from poultry litter had a high potential for supplying increased quantities of P and K to soils than biochar from lignocellulosic substrates [89]. Gao et al. (2018) [90] indicated that only biochar combined with NPK fertilizers can increase the bioavailability of phosphorus in soil. Brtnicky et al. (2023) concluded that the application of biochar with and without a mineral fertilizer can increase the microbial activity and fertility in the investigated soil, but leaching of nutrients from fertilizers could be mitigated by activation of biochar with a fertilizer [91]. Another study demonstrated that in an indoor simulation experiment, with increasing quantities of biochar, parameters such as soil pH, total nitrogen, available nitrogen, total phosphorus, and available phosphorus also increased. However, the researchers observed that the leaching solution from this experiment decreased. They concluded that when 2 kg·m^−2^ of biochar was added to soil, the cumulative leaching losses of total nitrogen, total phosphorus, ammonia nitrogen, and nitrate nitrogen were reduced by 13.57%, 26.09%, 13.9%, and 29.79%, respectively [92]. 

## 4. Discussion

Replacing peat with various alternatives has been a tremendous challenge for horticulture now. A wide range of substrates have been considered alternatives to phase out this contentious input from horticulture. This also includes biochars obtained from a variety of materials at different process parameters. Despite the fact that different types of biochars and their applications have been extensively studied, still there are very few reports demonstrating the effect of biochars for peat replacement in horticulture. In addition, these reports do not provide sufficient information on poultry-manure-derived biochars and their properties for peat substitution in growing media, such as the concentration of contaminants, bulk density, and water holding capacity. There is also insufficient information on differences in properties between biochars obtained from plant biomass vs. manure (e.g., poultry manure) (Table 1). 

Generally, depending on the pyrolysis temperature and the type of poultry manure, biochar is characterized by the following parameters: pH 7.20–10.5, C 33–86%, H 0.3–5.6%, N 0.12–4.9%, S 0.4–3.5%, O 0.01–42%, Na 1.5–2.9%, P 1–9.1%, and CEC 29–86 cmol·kg^−1^ [25,67,93]. The elemental composition of the biochar, specific surface area, pH, porosity and nutrient content, and stability and function of the surface groups can be modified by the temperature of the process [26,67,94]. Several researchers [30,93] have confirmed that the higher the process temperature was, the more ash was present in biochar. Consequently, high content of ash increases the pH of the obtained biochar. The decrease in the yield of biochar from poultry manure with increasing pyrolysis temperature was also reported by Bavariani et al. (2019) [25] and Sobik-Szołtysek et al. (2021) [30]. The yield of biochar production depends mainly on the used type of pyrolysis substrate. From the available data, it could be observed that the yield of animal-manure-derived biochars in some cases is higher than from plant-derived biochars produced at the same temperatures (Table 9). For all substrates, the biochar yield decreased with the increase in pyrolysis temperature. Biochars contain different elements such as Ca, Mg, P, and K. They can also contain various contaminants such as heavy metals, furans, dioxins, polychlorinated biphenyls, and polycyclic aromatic hydrocarbons, and thus biochars have to fulfill the requirements of the permissible limits if applied to the soil [95,96]. Bulk density and water holding capacity of biochars used as peat alternatives have hardly been reported in the studies on peat substitution by biochars. These properties are important to assess the potentials for different materials to be used as peat alternatives. 

The quality requirements for biochar input into the soil include the content of heavy metals (As, Cd, Cr, Cu, Pb, Hg, Ni, and Zn), polycyclic aromatic hydrocarbons (PAH-16), polychlorinated biphenyls (PCB-7), furans and dioxins (PCDD/F), dry matter content, pH, total organic carbon (TOC), total nitrogen (N) and potassium (K), phosphorus (as P_2_O_5_), and total calcium (Ca) and magnesium (Mg). According to the EBC standard, the biochar should have more than 60% of moisture content, N from 6 to 10%, TOC > 20%, Cd not more than 1.5 mg·kg^−1^, Cr < 100 mg·kg^−1^, Cu < 200 mg·kg^−1^, Ni < 50 mg·kg^−1^, Pb < 120 mg·kg^−1^, Zn < 600 mg·kg^−1^ [71]. There are no specific requirements for materials, including biochars, which could be used as peat alternatives in growing media. 

To the best of our knowledge, no studies on the use of poultry-manure-derived biochars, specifically to substitute peat in horticulture and their effect on the properties of growing media and plant growth, could be found in the literature. However, from the assessment of the properties of various poultry-manure-derived biochars reported in the literature and the presented study, these biochars demonstrate potentials for substituting peat in growing media for horticultural purposes. The properties shown by poultry-manure-derived biochars can make them suitable for partial or complete peat substitutes.

There are many advantages of using biochars produced from poultry manure to phase out peat from horticulture. It is argued that thermal conversion of poultry manure to biochar could be a solution for managing and handling (i.e., transportation, storage, processing, etc.) the excessive quantities of poultry manure. Converting excessive quantities of poultry manure through pyrolysis into biochar can reduce potentially toxic elements, destroy pathogenic microorganisms, and reduce the emissions of greenhouse gasses [84,107]. Thus, poultry-manure-derived biochars could be considered a sustainable and circular alternative to peat. What is more, the PMBs demonstrated required stability (for PMBs obtained at 525 °C, 625 °C, and 725 °C the H/C_org_ less than 0.7). The properties of these biochars can be engineered by adjusting pyrolysis parameters and using physical/chemical modifications. In our study, we confirmed that poultry-manure-derived biochars produced at higher temperatures are considered safe and thus do not pose a risk when they are added to soil. The concentrations of heavy metals (except from Cr in all investigated PMBs), dioxins and furans, PAHs, and PCBs are within the permissible limits and therefore can be used safely in horticulture. The investigated biochars showed the ability to release nitrogen through leaching. Biochar produced at 525 °C demonstrated a high nitrogen release effect. Due to high pH values (pH in the range of 9.24–12.35), biochars from poultry manure can have a liming effect when added as a peat substitute to the growing media. High water holding capacity of biochars—although still subject to research—could maintain moisture content in the growing media, and when added to soil, could increase the water availability to plants [69]. 

Despite the number of advantages, there are also some limitations to the use of poultry-manure-derived biochars for peat substitution. These biochars contain low contents of organic carbon (depending on the pyrolysis temperature), ranging from 29% to 36%, and thus this is considered a limitation for using biochar in horticulture. Low concentration of soluble phosphorus in biochars can result in lower availability. The literature reports that the availability of phosphorus in biochar can be estimated at 15% [108]. The concentrations of some heavy metals, e.g., Cr or Cu in poultry manure biochars, can be attributed to the fact that some of the metals are present in higher concentrations in animal manures [109]. 

## 5. Conclusions

In view of the presented results on the selected physico-chemical and physical properties, poultry-manure-derived biochars demonstrated potential to be used as substrates to phase out peat in growing media. The investigated biochars can have a liming effect if used as a partial substitution of peat. Water holding capacity of these biochars could influence water availability for plants when added to soil. Due to the content of P, K, Ca, and Mg, they can increase the content of these elements in the growing media. The biochars produced from poultry manure do not pose the risk related to the contents of heavy metals (except for Cr), dioxins and furans, and PAHs and PCBs. The pyrolysis temperature can be used to engineer some of the biochar properties.

More research work is needed to learn how the addition of poultry-manure-derived biochar for peat replacement will affect selected soil parameters such as pH and water holding capacity, alongside the content of nutrients in growing media, as well as have an impact on seed germination and plant growth. Future research aims at testing poultry-manure-derived biochar with other wood-derived substrates and compost as partial substitution of peat in growing media and to compare it with commercially available peat-based growing media.

## Figures and Tables

**Table 1 materials-16-06392-t001:** Properties of poultry-manure-derived biochars reported in the literature.

Temp. °C	pH	Ash %	N %	TC %	Ca g·kg^−1^	Mg g·kg^−1^	P g·kg^−1^	K g·kg^−1^	BET m^2^·g^−1^	BD kg·m^−3^	WHC %	CEC cmol (+) kg^−1^	References
200	7.20	-	3.53	39.7	-	-	3.39	1.04	-	-	-	580	[25]
300	7.30	-	3.80	42.4	-	-	4.13	1.26	-	-	-	690	
300	8.00	36.50	3.52	39.07	-	-	-	-	4.00	-	-	-	[26]
300	9.68	40.09	4.30	40.47	0.07	0.03	39.20	5.85	4.51	-	-	-	[27]
300	9.5	47.87	4.91	37.99	23.88	0.28	16.59	32.01	2.8	-	64.32	52	[28]
350	9.9	-	-	-	-	-	-	-	4.00	-	-	-	[29]
350	10.3	51.29	3.49	37.65	22.57	0.13	13.33	34.18	3.5	-	59.56	45	[28]
400	10.4	56.62	1.46	36.10	13.08	0.07	7.00	36.67	4.0	-	52.38	40	[28]
400	9.98	-	4.70	47.9	-	-	5.58	1.72	-	-	-	750	[25]
425	10.40	52.07	4.81	37.98	12.70	1.32	3.65	4.93	12	-	-	31.9	[30]
450	10.00	-	-	-	-	-	-	-	7.00	-	-	-	[29]
450	10.5	58.66	1.15	35.22	9.56	0.06	5.07	39.17	4.5	-	48.30	38	[28]
475	12.04	50.20	3.73	30.76	14.69	1.00	19.27	3.24	-	-	-	-	[31]
500	11.3	60.58	1.12	34.47	9.18	0.05	4.44	40.40	5.0	-	46.18	35	[28]
500	11.02	50.00	3.15	34.41	0.05	0.11	45.93	6.40	8.08	-	-	-	[27]
500	11.50	-	4.50	55.1	-	-	6.38	1.97	-	-	-	865	[25]
500	9.1	-	2.13	29.67	54.00	5.1	19.4	17.2	11.51	-	60	35.59	[32]
550	10.20	-	-	-	-	-	-	-	6.00	-	-	-	[29]
550	11.00	60.65	1.25	33.88	8.54	0.05	4.15	43.89	5.5	-	44.47	32	[28]
550	7.69	46.20	3.81	33.7	-	-	-	-	6.97	-	-	222	[33]
525	10.65	61.74	2.50	29.00	16.30	1.41	3.28	4.47	17	-	-	118.9	[30]
580	7.56	8.21	0.65	52.3	0.75	0.26	0.73	1.25	-	-	-	-	[34]
580	7.86	8.32	0.85	55.7	4.63	0.07	0.38	1.92	-	<0.1	-	-	[35]
600	11.50	60.78	1.33	32.52	8.24	0.05	2.82	44.61	6.0	-	41.85	30	[28]
600	9.22	49.99	1.86	32.30	-	-	-	-	86.67	-	-	-	[26]
675	12.55	-	3.07	30.56	14.03	1.00	17.23	3.01	-	-	-	-	[31]
680	10.1	11.16	1.3	86.79	-	-	-	-	6.96	-	-	-	[36]
700	11.81	54.78	2.84	33.77	0.15	0.21	49.51	6.39	10.89	-	-	-	[27]
725	12.45	78.38	2.76	37.42	18.1	1.50	4.00	5.55	19	-	-	386.3	[30]
775	13.40	-	3.69	30.29	14.87	0.93	15.46	2.66	-	-	-	-	[31]
800	12.2	68.2	2.2	23.9	-	-	-	-	-	-	-	-	[37]
800	10.11	64.63	2.01	30.35	-	-	-	-	-	-	-	-	[38]

**Table 2 materials-16-06392-t002:** Biochar yields at the selected pyrolysis temperatures.

	425 °C	525 °C	625 °C	725 °C
Biochar yield, %	51.24 ± 0.77	47.43 ± 0.45	44.41 ± 0.51	39.59 ± 0.82

**Table 3 materials-16-06392-t003:** Selected physico-chemical and physical properties of the obtained biochars.

	PMB 425 °C	PMB 525 °C	PMB 625 °C	PMB 725 °C
pH_H2O_	9.24	10.18	11.10	12.35
Ash, % (d.m.)	55.61	63.91	63.50	73.34
Total carbon (TC), % (d.m.)	38.57	37.70	38.20	31.28
Ca, % (d.m.)	12.50	12.20	15.10	16.70
Mg, % (d.m.)	0.83	1.45	1.43	1.53
P, % (d.m.)	2.03	3.91	3.70	3.68
K, % (d.m.)	2.74	4.96	4.76	5.13
Brunauer–Emmett–Teller (BET) surface area, m^2^·g^−1^	12	13	11	18
t-plot micropore area, m^2^·g^−1^	5.3	3.2	2.5	2.6
Cation exchange capacity (CEC), cmol (+)·kg^−1^	136.2	111.7	226.1	481.5
Bulk density (BD wet), kg·m^−3^	200	199	182	251
Water holding capacity (WHC), % *w*/*w*	158	219	217	232

**Table 4 materials-16-06392-t004:** Elemental analysis of the obtained biochars.

% (d.m)	PMB 425 °C	PMB 525 °C	PMB 625 °C	PMB 725 °C
TOC	29.30	29.16	35.78	32.47
H	2.37	1.56	1.17	1.36
N	4.00	3.26	3.03	3.02
S	0.30	0.52	0.49	0.51
H/C_org_	0.89	0.57	0.42	0.56

**Table 5 materials-16-06392-t005:** The contents of heavy metals in the obtained biochars.

Heavy Metals mg·kg^−1^ (d.m.)	Permissible Limits according to the Regulation (EU) 1009/2019	PMB 425 °C	PMB 525 °C	PMB 625 °C	PMB 725 °C
Cd	2	0.580	0.610	<0.300	<0.300
Pb	120	<2	2.05	<2	2.02
Cr	2	17.10	29.80	29.60	24.60
Cu	300	84.6	156	158	144
Ni	50	16.4	26.6	28.0	32.8
Zn	800	434	831	775	747
Hg	1	<0.0006	<0.0006	0.0008	0.0007

**Table 6 materials-16-06392-t006:** Dioxins and furans in the obtained biochars.

Type of Contaminant ng·kg^−1^ (d.m.)	PMB 425 °C	PMB 525 °C	PMB625 °C	PMB 725 °C
2378-TCDD	<9.20	<1.10	<1.10	<1.10
12378-PeCDD	<1.40	<1.30	<1.20	<1.30
123478-HxCDD	<1.80	<1.60	<1.70	<1.70
123678-HxCDD	<1.80	<1.60	<1.70	<1.70
123789-HxCDD	<1.80	<1.60	<1.70	<1.70
1234678-HpCDD	<1.90	<1.80	<2.20	<2.30
OCDD	<9.30	<2.10	<2.70	<3.20
2378-TCDF	<1.20	<1.10	<1.20	<1.10
12378-PeCDF	<1.50	<1.40	<1.40	<1.30
23478-PeCDF	<1.50	<1.40	<1.40	<1.30
123478-HxCDF	<1.90	<1.80	<1.90	<1.80
123678-HxCDF	<1.90	<1.80	<1.90	<1.80
123789-HxCDF	<1.90	<1.80	<1.90	<1.80
234678-HxCDF	<1.90	<1.80	<1.90	<1.80
1234678-HpCDF	<2.20	<1.90	<2.30	<2.50
1234789-HpCDF	<2.20	<1.90	<2.30	<2.50
OCDF	<2.40	<2.30	<3.50	<2.50
TEQ-Lowerboud	0.00	0.00	0.00	0.00
TEQ-Upperbound	3.90	3.80	3.90	3.90

**Table 7 materials-16-06392-t007:** PAHs and PCBs in the obtained biochars.

Type of Contaminant mg·kg^−1^ (d.m.)	PMB 425 °C	PMB 525 °C	PMB 625 °C	PMB 725 °C
Naphthalene	0.097	0.089	1.460	16.700
Acenaphthylene	<0.010	<0.010	0.214	1.430
Acenaphthene	<0.010	<0.010	0.052	0.748
Fluorene	0.032	<0.010	<0.040	<0.220
Phenanthrene	0.035	0.036	0.207	3.160
Anthracene	0.010	<0.010	0.045	<1.090
Fluoranthene	<0.010	<0.010	0.050	<0.560
Pyrene	<0.010	<0.010	0.070	0.647
Benzo(a)anthracene	<0.010	<0.010	<0.030	- *
Chrysen	0.016	<0.010	<0.011	- *
Benzo(b)fluoranthene	<0.010	<0.010	0.012	- *
Benzo(k)fluoranthene	<0.010	<0.010	<0.011	- *
Benzo(a)pyrene	<0.010	<0.010	<0.011	- *
Indeno(1.2.3.cd)pyrene	<0.010	<0.010	<0.030	- *
Benzo(g.h.i)perylene	<0.010	<0.010	- *	- *
Dibenzo(a.h)anthracene	<0.010	<0.010	<0.030	- *
Sum of 16 PAHs	0.190	<0.160	- *	- *
PCB 28	<0.0030	<0.0030	<0.0030	<0.1290
PCB 52	<0.0030	<0.0030	<0.0030	<0.0030
PCB 101	<0.0030	<0.0030	<0.0030	<0.1140
PCB 118	<0.0030	<0.0030	<0.0030	<0.0210
PCB 138	<0.0030	<0.0030	<0.0030	<0.0030
PCB 153	<0.0020	<0.0020	<0.0020	<0.0020
PCB 180	<0.0030	<0.0030	<0.0030	<0.2550
Sum of 6 PCBs	<0.0170	<0.0170	<0.0170	<0.5060
Sum of 7 PCB	<0.0200	<0.0200	<0.0200	<0.5270

* Not determined due to the sample matrix complexity.

**Table 8 materials-16-06392-t008:** Sorption of N_NH4_ and P_PO4_ in the solution by the PMBs.

Type of Biochar	N_NH4_, % (the Solution Containing N_NH4_)	P_PO4_, % (the Solution Containing P_PO4_)	N_NH4_ i P_PO4_, % (the Solution Containing Both N_NH4_ + P_PO4_)	
			N_NH4_, %	P_PO4_, %
**PMB 425 °C**	−8.99 *	97.91	−9.81 *	81.27
**PMB 525 °C**	−62.26 *	97.20	−46.05 *	91.54
**PMB 625 °C**	−2.45 *	77.72	−2.45 *	71.11
**PMB 725 °C**	4.66	99.33	6.27	98.52

* Negative values indicate higher concentrations of nitrogen in the solution in comparison to the initial concentration of nitrogen in the solution (desorption).

**Table 9 materials-16-06392-t009:** Properties of various biochars used for peat substitution reported in the literature.

Type of Biochar	Temp. °C	pH	Ash %	N %	TC %	Ca	Mg	P	K	BET m^2^·g^−1^	BD g·m^−3^	WHC %	CEC cmol (+) kg^−1^	Yield %	References
**Plant derived**															
Peanut hull	400	7.6	8.2	2.7	74.8	-	-	0.04	0.15	-	-	-	2.4	-	[97]
Peanut hull	500	7.8	9.3	2.7	81.8	-	-	0.03	0.17	-	-	-	2.1	-	
Pecan shell	350	6.3	2.4	0.26	64.5	-	-	0.02	0.06	-	-	-	3.1	-	
Pecan shell	700	7.8	5.2	0.51	91.2	-	-	0.03	0.04	-	-	-	2.9	-	
Switchgrass	250	6.2	2.6	0.43	55.3	-	-	0.03	0.06	-	-	-	2.5	-	
Switchgrass	500	7.0	7.8	1.07	84.4	-	-	0.03	0.08	-	-	-	2.3	-	
Hardwood wastes	500	6.6	8.9	3.0	71.4	-	-	0.02	0.06	-	-	-	2.3	-	
Rice straw	500	10.7	27.9	1.18	47.22	5.63	9.94	1.05	0.47	1.89	-	-	23.1	37.6	[98]
Rice straw	700	11.1	31.8	0.83	41.65	6.33	6.56	1.31	0.6	199	-	-	20.0	33.7	
Wheat straw	500	9.2	14.5	0.53	49.78	0.90	0.31	0.70	24.0	2.48	-	-	20.1	30.6	
Wheat straw	700	10.7	17.9	0.34	52.59	0.94	0.31	0.73	25.7	319	-	-	15.4	26.8	
Maize straw	500	10.3	30.0	2.18	48.81	1.03	0.41	7.00	16.6	0.17	-	-	26.1	32.3	
Maize straw	700	10.7	29.4	1.97	51.12	1.12	0.49	7.96	18.8	20.5	-	-	24.8	26.1	
Rice husk	500	10.0	37.9	1.09	37.16	0.31	0.11	2.10	2.74	12.2	-	-	11.7	37.6	
Rice husk	700	10.8	40.4	0.92	37.51	0.29	0.13	2.40	2.65	136	-	-	7.4	33.7	
Cocount shell	500	10.3	7.3	0.35	56.10	0.33	0.21	0.72	20.4	69.4	-	-	31.1	32.7	
Cocount shell	700	10.8	8.7	0.21	56.00	0.28	0.18	1.31	24.3	341	-	-	20.2	27.8	
Bamboo wood	500	10.0	2.3	0.34	37.77	0.12	0.27	0.39	17.1	169	-	-	3.0	25.3	
Bamboo wood	700	8.9	2.5	0.26	38.59	0.25	0.14	0.30	2.34	306	-	-	0.5	23.0	
Elm wood	500	7.8	1.1	0.16	45.92	0.26	0.06	0.19	4.97	84.3	-	-	1.7	29.3	
Elm wood	700	9.2	1.8	0.03	47.04	0.44	0.13	0.16	3.71	325	-	-	0.1	21.2	
Hemlock	500	7.4	-	-	-	46	11	6	4	-	-	-	0.9	-	[99]
Switchblade grass	500	9.3	-	-	-	64	14	22	50	-	-	-	1.4	-	
Palm waste	600	-	13.86	0.93	67.28	-	-	-	-	3.28	-	-	-	26	[100]
**Manure derived**															
Dairy manure	100	8.0	95	3.12	36.8	3.23	1.11	0.91	-	2.0	-	-	-	38.1	[101]
Dairy manure	200	7.0	58.0	2.98	31.1	6.09	1.68	1.74	-	2.8	-	-	-	45.0	
Dairy manure	350	10.0	25.0	2.22	25.2	8.89	2.65	2.41	-	7.1	-	-	-	61.1	
Dairy manure	500	10.1	12.1	0.04	1.67	9.75	3.02	2.66	-	12.0	-	-	-	83.2	
Swine manure	400	7.5	43.5	3.2	1.0	5.5	3.0	6.1	3.1	5.7	-	-	-	31.0	[102]
Swine manure	500	10.2	45.8	2.6	1.0	5.7	3.4	6.9	2.7	3.9	-	-	-	32.1	
Swine manure	700	11.8	52.8	2.0	0.9	5.0	3.4	7.5	2.7	59	-	-	-	35.3	
Swine manure	800	11.4	51.8	1.6	1.1	5.3	3.4	7.7	2.7	63	-	-	-	30.2	
Yak manure	300	7.8	-	3.2	41.6	5.67	1.98	4.52	2.78	3.6	-	-	-	20.6	[103]
Yak manure	500	10.2	-	3.0	41.3	6.13	2.34	5.41	2.85	17.3	-	-	-	23.9	
Cow manure	300	8.3	-	1.7	51.3	-	-	1.36	1.3	-	-	-	-	58.07	[104]
Cow manure	500	10.6	-	1.45	52.54	-	-	6.12	3.5	-	-	-	-	39.84	
Cow manure	700	10.5	-	1.06	52.85	-	-	1.68	4.4	-	-	-	-	37.12	
Cow manure	500	9.2	-	1.51	33.61	2.12	1.4	8.14	0.14	-	-	-	4.84	-	[105]
Pig manure	300	7.8	50.25	2.24	-	1.3	0.8	2.8	0.8	-	-	-	35.6	-	[106]
Pig manure	500	8.2	73.88	1.19	-	0.8	0.8	1.2	0.8	-	-	-	32.7	-	

## Data Availability

Not applicable.

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
