# Peer review of "Properties of Poultry-Manure-Derived Biochar for Peat Substitution in Growing Media"

_materials, 2023, doi:10.3390/ma16196392_

Round 1
Reviewer 1 Report
Journal : materials
Title: Properties of poultry manure-derived biochar for peat substitution in growing media
Authors: Katarzyna Wystalska1, Krystyna Malińska1*, Jolanta Sobik-Szołtysek1, Danuta Dróżdż 1 and Erik Meers 2
Ms No. materials-259854
Here, poultry manure derived biochar (PMB) was prepared as an alternative for peat. The scope included: laboratory scale pyrolysis of poultry manure at the temperatures of 425-725°C, determination of selected physical and chemical properties of the obtained biochars, including contaminants, and assessment of the potentials of produced biochars to be used as peat substitutes.
Peat is considered a contentious input in horticulture. For example, even in China, much peat is bought from northern Europe. Excessive utilization of peat would result in the negative effects to the environment. Thus, substitution should be considered.
Biochar is a new material, causing much interesting due to its utilization in climate change, carbon sink, waste water treatment, crop growth and soil contamination control. However, risk would occur in the biochar application because there are HMs and PAHs etc. contaminants in biochar, limiting its usage.
Thus, the topic of this paper is very interesting, the data were plentiful and the comparison with other biochars was done. The reviewer thinks the results would useful in practice.
The paper could be accepted after a minor revision (A concise introduction is expected).
Author Response
Thank you so much for the overall positive feedback on our manuscript. The authors share the same concerns and hopes for biochar as a peat substitute. Thank you for commenting on the introduction part which indeed is extensive. This was our intention to justify in detail the need for alternative materials for phasing out peat from horticulture and to demonstrate that despite a great number of publications on biochar, in particular biochar produced from poultry manure, the data on biochar properties - which are important from the perspective of substituting peat with biochar - are limited. It might be difficult to compare the results from different studies as in many cases there is a lack of information on substrates (e.g. type of poultry manure), pretreatments, process parameters and properties of biochars (e.g. particle size, bulk density, porosity, water holding capacity, sorption properties, etc.). Therefore, we would like to keep the introduction in the present form, as in our opinion it gives the reader better understanding of the problem and how we are approaching to find the solution to it.
Reviewer 2 Report
1. Explain the mechanism for how the properties were obtained during the thermochemical process
2. The bulk density, BET surface area, and microspore area could explain the water holding capacity and absorption of N and P. Discuss how different biochar pyrolysis temperatures influence those factors
3. Discuss the effect of pyrolysis temperature on the toxic substrate generation
4. Root could be sensitive to pH; if possible, discuss biochar pH for the root growth
5. As the biochar ash content is high, it will be better if the effect of ash is discussed
6. Adding some references that show manure biochar for plant growth might be good. https://doi.org/10.5539/JAS.V11N4P515 , https://doi.org/10.4025/ACTASCIAGRON.V35I3.17542 etc.
7. Check the latest version of IBI, “Biochar Standardized Product Definition and Product Testing Guidelines for Biochar That Is Used in Soil.” Appendix 3 can incorporate the table 6
8. If other peat-substitute growing media are compared to biochar , it will be informative
9. Consider germination tests to assess biochar quality (1-2 months)
10. All abbreviations should be explained the first time it is mentioned. 11. Explain all abbreviations used in the table.
12. The format of decimal should be consistent ( . or ,) (table 7 and 8 are using , )
13. So many tables, if possible, reduce the number or plot it.
Author Response
Thank you so much for your valuable feedback on our study which we used to improve the manuscript. We really appreciate your time and effort to analyze and advise on the issues addressed in your comments. As suggested we introduced the changes you suggested to improve the manuscript.
Below, we tried to explain and clarify the aspects you addressed in your comments.
Response to #1
Thank you for this comment. In general, biochars obtained from different substrates (plant and animal derived biochars) and processed through pyrolysis at various process parameters such as pyrolysis temperature, heating time, holding time, etc. demonstrate a wide variety of properties. Adjusting process parameters can allow to some extent to tailor the properties of biochars in view to selected applications. The temperature plays a crucial role in tailoring the properties of biochars as it affects physical and chemical properties such as chemical composition, specific surface area, types and presence of functional groups, porosity, etc. In addition, properties of biochars can be tailored or adjusted by chemical or thermal modifications. In our opinion, this is a very complex issue and substrate/biochar specific, and such discussion supported with more data could be addressed in a separate manuscript.
We addressed how pyrolysis, in particular the temperature, affects the selected properties of biochar in section 3.2 but added more information as suggested by the reviewer. Also, we addressed the impact of pyrolysis temperature on selected properties in Results and Discussion sections but mostly in view of poultry manure-derived biochars and selected temperatures (425-725C).
Response to #2
Thank you so much for this comment. We already provided some explanation for the effect on pyrolysis temperatures on bulk density, BET and microspore area in the Results section but as suggested by the reviewer we added more information in the text.
Response to #3
Thank you for this comment. We added more information on this effect in the Introduction part as well as in the Results and Discussion parts providing the relevant and recent references. This includes a very interesting study reported by Sormo E., Krahn K.M., Flatabo G.O., Hartnik T., Arp H.P.H. Cornelissen G. Distribution of PAHs, PCBs, and PCDD/Fs in products from full-scale relevant pyrolysis of diverse contaminated organic waste. Journal of Hazardous Materials 132546, 2023 https://doi.org/10.1016/j.hazmat.2023.132546
Response to #4
Thank you very much for the comment. Yes, we agree that plant roots could be sensitive to pH. In this study we focused on physico-chemical and physical properties of biochars from poultry manure. We did not perform any phytotoxicity tests with the produced biochars but this is planned for our next study. However, the literature reports that very few studies on the impact of poultry manure-derived biochar pH on root systems. In the study by Solaiman, Z. M., Shafi, M. I., Beamont, E., Anawar, H. M. Poultry litter biochar increases mycorrhizal colonisation, soil fertility and cucumber yield in a fertigation system on sandy soil. Agriculture 2020, 10(10), 480. https://doi.org/10.3390/agriculture10100480 the authors stated that biochar with pH above 8.0 has a beneficial effect on plant growth when acidic soils are considered. In our case in would be difficult do discuss the effect of pH of the produced biochars on the growth of plant roots as we have not performed the experiment yet to support such discussion.
Response to #5
Thank you for this comment. Biochars produced from animal derived feedstock demonstrate high ash content. Similarly, the investigated biochars from poultry manure demonstrated the ash content ranging from about 56% to 73% in the temperatures of 425-725C. The highest ash content was reported for the temperature of 725C which is due to the significant loss of volatile substances. We addressed this issue in the text. Despite the fact that ash content is high in biochars produced from animal feedstock, still the fertilizing potential of such biochars is investigated.
Response to #6
Thank you so much for recommending these publications. We found them very interesting. However, the first one (i.e. Substrates Formulated with Biochar for Seedling Production of Moringa oleifera Lam by Soares et al., 2019) addresses 3 types of biochar produced from dry coconut shells, sewage sludge and orange bagasse. This study does not provide any information on poultry manure derived biochar. Neither does the other paper the reviewer was recommending (Biochar as substitute for organic matter in the composition of substrates for seedlings by Lima et al., 2013). Both are very interesting and we would refer to them in other work.
Response to #7
Thank you so much for this suggestion. In this study we followed the requirements for soil enhancers provided by the Fertilizing Product Directive, i.e. Regulation 1009/2019 available at ]>Regulation (EU) 2019/ of the European Parliament and of the Council of 5 June 2019 laying down rules on the making available on the market of EU fertilising products and amending Regulations (EC) No 1069/2009 and (EC) No 1107/2009 and repealing Regulation (EC) No 2003/2003 (europa.eu)
and also ANNEXES to the Commission Delegated Regulation amending Annexes II, III and IV to Regulation (EU) 2019/1009 of the European Parliament and of the Council for the purpose of adding pyrolysis or gasification materials as a component material category in EU fertilising products.
We are familiar with the voluntary standards of the IBI and EBC and perhaps it would be very informative to include the table with recommended parameters for biochars but we decided not to include this in the submitted manuscript as we are already presenting a lot of data in the tables.
Response to #8
Thank you so much for this comment. Indeed, this would be interesting and informative to compare other feedstock materials with biochar and peat to have a better understanding of the properties of these materials and their potentials for phasing out peat from horticulture. It was not our aim for this study to compare other materials which could be used as alternatives to peat. But this is an idea for a review paper. Thank you for suggesting this.
Response to #9
Thank you so much for this suggestion. This is what we are planning to do next. The manuscript presents just the physical and physcio-chemical properties of poultry manure-derived biochars that are important if considered for peat substitution.
Response to #10
Thank you for pointing this out. We explained them when first used. This includes the abstract as well.
Response to #11
Thank you for pointing this out. We included full names of the abbreviations in the tables for clarity and convenience for the reader.
Response to #12
Thank you for pointing this out. We made the corrections.
Response to #13
Yes, indeed, we are presenting the results with 10 tables (2 of them are based on the outcome from reviewing other studies). Our intention was to clearly present the data sets and we followed the Reviewer’s advice. We managed to combine table 3 and 4 into one (table 3).
Reviewer 3 Report
Comments for Editor and Authors
In the present study the authors prepared biochar by the pyrolysis of poultry manure at the temperatures of 425-725°C, determined the selected physical and chemical properties of the obtained biochars, including contaminants, and assessed the potentials of produced biochars to be used as peat substitutes. The study is well planned and the results obtained were interpreted by examining the experimental parameters in detail. In my opinion the manuscript may be accepted for publication in Materials after considering the major comments given below;
1. Has the amount of nitrogen in the original sample been determined? If determined, was there any nitrogen loss during the biochar production process? If so, isn't this a disadvantage in terms of alternative fertilizer? What was the initial amount of N, P and other elements in the original raw material on a dry basis? Was there any nitrogen or phosphorus loss during biochar production? Was there partial loss of NOx and POx, even in a limited oxygen environment? Could this situation be a disadvantage in terms of developing alternative fertilizers?
2. According to Table 9, if N adsorption is negative, it means that the fertilizer releases nitrogen even when there are nitrogenous compounds in the soil. This is a good feature. However, what does the fact that fertilizer adsorbs phosphate in the same environment mean in terms of the efficiency of fertilizer? This is probably not a desired situation in terms of fertilizer efficiency.
3. In line 162, there is no analysis method called ICP. Is this either ICP-OES or ICP-MS?
4. The whole name of the abbreviations (such as TOC, WHC, etc.) given in the abstract section should be written first and then abbreviated.
5. It would be more meaningful if it is written which poultry manure is used in each reference in Table 1.
6. In line 139-141, Are the analysis results given at these lines belong to the current study? Or is it literature data? This part is not understood. If the results obtained in this study, they should be moved to the Results Discussion section.
7. In line 181, The reason for adjusting the pH value to 4.9 in NNH4 and PPO4 sorption experiments should be stated.
8. In Table 7, the second column, where units are given, should be removed from the table. Instead of giving units in each row, the table caption should be written as " Dioxins and furans levels in dry biochars (mg/kg)”.
9. In Table 8, the second column, where units are given, should be removed from the table. Instead of giving units in each row, the table caption should be written as " PAHs and PCBs levels in dry biochars (mg/kg)”.
10. In tables, a full stop should be used instead of a comma as a decimal separator.

Author Response
Thank you so much for your valuable feedback on our study which we used to improve the manuscript. We really appreciate your time and effort to analyze and advise on the issues addressed in your comments. As suggested we introduced the changes you suggested to improve the manuscript.
Below, we tried to explain and clarify the aspects you addressed in your comments.
Response to #1
Thank you so much for this comment. These are very important issues to be addressed if biochar is considered to be developed into a bio-based fertilizer or a peat substitute. In this study we were looking specifically into physico-chemical and physical properties of poultry manure-derived biochars produced at different temperatures and assessed their potentials for partial or complete substitution of peat to phasing out peat from horticulture. We did not aim to analyze the loss of nitrogen and phosphorus during conversion of poultry manure through pyrolysis at selected temperatures into biochars and we did not plan to include this analysis in the scope of this work. The properties of poultry manure used in our experiments, i.e. nitrogen and carbon contents, are included in section 2.1. We also added more information on the composition of fresh poultry manure, as suggested by the reviewer. However, converting poultry manure into biochar results in the loss in nitrogen. According to our results half of the total nitrogen content was lost in the process. This is due to thermal degradation of nitrogen compounds. Also, the nitrogen loss during pyrolysis is higher with higher pyrolysis temperature which was demonstrated in this study. As for phosphorus, according to the results from our study, there is no loss of phosphorus. The content of phosphorus increased with higher pyrolysis temperatures.
Response to #2
Thank you so much for this comment. We are also sharing your concern about the efficiency of poultry manure-derived biochar as a fertilizer. We are planning additional experiments on sorption properties of poultry manure-derived biochars with special focus on phosphorus sorption properties. Our aim was to investigate adsorption of the excess phosphorus. We look at this phenomenon as the ability of poultry manure-derived biochar to bind phosphorus, and thus to function as a buffer for this element in the soil environment. This might be considered important as the excess of phosphorous is linked to deficiency of such microelements like Fe and Zn, and in consequence this can affect the plant metabolism
Response to #3
Thank you for pointing this out. It is ICP-OES. We added the missing part.
Response to #4
Thank you for pointing this out. We introduced the full names of the abbreviations, including in the abstract section.
Response to #5
In Table 1 we included the available data on the properties of poultry manure-derived biochars with corresponding references (the last column in the table). The literature references hardly provided specific information on the type of poultry manure used for obtaining biochar. As temperature is a crucial parameter we organized the table listing poultry manure-derived biochars obtained at specific temperature. This is one of our conclusions that despite a significant number of publications on poultry manure-derived biochars often times it is difficult to compare the results of the published studies as not all parameters were determined.
Response to #6
Thank you so much for pointing this out. In the sentence you are referring to:
“On average, the moisture content was about 80%, organic matter – 75% (d.m.), organic carbon – 43% (d.m.), nitrogen – 8% (d.m.) and pH – 7.5. Bulk density (wet) was about 910 kg m-3 whereas air-filled porosity was 20% [30,31,39].”
we provided the selected parameters of poultry manure sampled from a poultry farm and used in our studies in the project Nutri2Cycle. This poultry manure was used as a substrate in our experiments which included composting, pyrolysis and anaerobic digestion. Therefore, we presented the properties of poultry manure in the section with Materials and Methods. We included the average values of selected parameters of poultry manure sampled throughout the project from the same poultry farm, the same type of breeding, the same type of laying hens, feeding regime, etc.). In addition, we provided the references to the publications (indicated in the reference list as 30, 31 and 39) in which we were also reporting the characteristics of poultry manure used as a substrate to obtain biochar or compost.
We clarified this by adding the explanation in the text.
Response to #7
Thank you for pointing this out. We added the explanation for adjusting the pH. The reason for adjusting pH to 4.9 was related to our studies on acidic soils (typical in Poland). The soil solutions demonstrated low pH around 4.9.
Response to #8
Thank you for suggesting this. We introduce this change to the table.
Response to #9
Thank you for suggesting this. We introduced this change to the table.
Response to #10
Thank you for pointing this out. We made the correction.
Round 2
Reviewer 3 Report
Dear Editor,
I have reviewed the manuscript entitled “Properties of poultry manure-derived biochar for peat substitution in growing media” once more. The authors have made an effort to improve the manuscript, and also they made most of the changes necessary. However some situations that are overlooked are given below. I recommend that this paper can be published in Materials after considering the following comments;
· In Table 6, it is recommended to write the concentration of contaminants as ng/kg instead of ng/g. In this case, the values in the table multiplied by 1000 may be more understandable to readers.
· In line 214 “Th” should be “The”. In line 357 “suraface” should be “surface”. In line 396 “intiital” should be “initial” “soltuion” should be “solution”. In line 475 “litrature” should be “literature”.
· In line 388, the reference 70 should be changed as 71.
Dear Editor,
I have reviewed the manuscript entitled “Properties of poultry manure-derived biochar for peat substitution in growing media” once more. The authors have made an effort to improve the manuscript, and also they made most of the changes necessary. However some situations that are overlooked are given below. I recommend that this paper can be published in Materials after considering the following comments;
· In Table 6, it is recommended to write the concentration of contaminants as ng/kg instead of ng/g. In this case, the values in the table multiplied by 1000 may be more understandable to readers.
· In line 214 “Th” should be “The”. In line 357 “suraface” should be “surface”. In line 396 “intiital” should be “initial” “soltuion” should be “solution”. In line 475 “litrature” should be “literature”.
· In line 388, the reference 70 should be changed as 71.
Author Response
Dear Rviewer,
thank you so much for your comments.
Regarding you comments:
Comment #1 In Table 6, it is recommended to write the concentration of contaminants as ng/kg instead of ng/g. In this case, the values in the table multiplied by 1000 may be more understandable to readers.
Thank you so much for this advice. We changes the numbers into ng/kg.
Comment #2 In line 214 “Th” should be “The”. In line 357 “suraface” should be “surface”. In line 396 “intiital” should be “initial” “soltuion” should be “solution”. In line 475 “litrature” should be “literature”.
Thank you for spotting these typoes. We corrected them.
Comment #3 In line 388, the reference 70 should be changed as 71.
Thank you for spotting this one too. We changed it.
Again, thank you so much for reviewing and helping us to improve the manuscript.
